# Telerobotic Intergroup Contact: Acceptance and Preferences in Israel and Palestine

**DOI:** 10.3390/bs14090854

**Published:** 2024-09-23

**Authors:** Avner Peled, Teemu Leinonen, Béatrice S. Hasler

**Affiliations:** 1Department of Art and Media, Aalto University, FI-00076 Aalto, Finland; teemu.leinonen@aalto.fi; 2Sammy Ofer School of Communications, Reichman University, Herzliya 46150, Israel; hasler.beatrice@gmail.com

**Keywords:** intergroup contact, intergroup communication, telerobotics, telepresence, Israeli–Palestinian conflict, human–robot interaction, human–computer interaction

## Abstract

We explore telerobotics as a novel form of intergroup communication. In this form, remotely operated robots facilitate embodied and situated intergroup contact between groups in conflict over long distances, potentially reducing prejudice and promoting positive social change. Based on previous conceptual frameworks and design hypotheses, we conducted a survey on the acceptance and preferences of the telerobotic medium in Israel and Palestine. We analyzed the responses using a mixed-method approach. The results shed light on differences in attitudes between the groups and design considerations for telerobots when used for intergroup contact. This study serves as a foundation for the implementation of a novel method of technology-enhanced conflict resolution in the field.

## 1. Introduction

Seventy years have passed since Gordon Allport [1] hypothesized that, under certain conditions, contact between individuals in opposing groups could reduce the amount of prejudice between them. These facilitating conditions are as follows: (a) equal status between the groups within the situation, (b) common goals, (c) cooperation between the groups, and (d) institutional support. Over the years, the research community has been refining and problematizing Allport’s broad, yet succinct observations into a full-fledged model of how we could make the world better by bringing people together.

A crucial development in intergroup communication research has been to conclude that bringing people together does not necessarily mean that they must be situated together in the same physical space [2]. Forms of indirect contact [3,4,5,6,7] emerged mainly as a response to concrete challenges in gathering people from conflicting groups at the same place, but have since then been considered as worthy in their own right, providing complementary benefits to face-to-face contact, such as reduced anxiety [8] and easier integration with public policy [9].

This research answers the call for a new generation of intergroup contact research [10] by proposing, to our knowledge, a so far unexplored medium for contact: telepresence robots, or more generally, telerobots. A recent meta-analysis by Imperato et al. [11] concluded that computer-mediated communication (CMC) has the potential to reduce intergroup prejudice. Online contact (or e-contact) has a positive effect, especially when it follows the original conditions of Allport, such as team collaboration and having an organized and structured encounter, as opposed to random conversations on social media, which are often negative and polarizing [12,13,14]. At the same time, it is evident that increasing the immediacy of communication (e.g., using video rather than text) promotes the perceived social presence of the interlocutors, increases intimacy in the conversation, and in turn produces positive effects [15]. Embodied and immersive virtual reality (VR) environments were also successful in promoting intergroup empathy in passive, indirect contact experiments [16,17,18,19]. However, as we have learned from the Zoom era of COVID-19, virtual conferencing is often not an adequate replacement for face-to-face contact [20,21,22].

To bring e-contact down to earth, we proposed the use of remote-controlled robots as a medium for intergroup communication [23,24]. In this form, an ingroup member remotely operates a robot using a device such as a mobile phone or a desktop computer and communicates via the robot with an outgroup member who is physically present with the robot. Initially, such contact could be realized by a third-party organization that places a robot in the physical location of the outgroup and invites members of the ingroup to operate it online and converse with the locals. As telerobots become commonplace and affordable, we could speculate that such encounters will occur organically. This form of contact enjoys a mix of benefits from immediate, situated face-to-face communication (between the outgroup member and the robot) and also from remote, virtual communication (mediated by the robot). Our previous research [23,24] outlines a conceptual framework and a set of hypotheses for telerobotic contact. The use of a telerobotic medium introduces new design affordances that may have an impact on intergroup communication. A robot could be equipped with physical and virtual accessories and features that determine its identity, agency, and how much it reveals about the identity of its teleoperator. In this study, we present a survey of user preferences and attitudes toward telerobotics, in accordance with our previous hypotheses. The following paragraphs outline the context of the survey and the hypotheses embedded in this study.

We choose to focus on the Israeli–Palestinian conflict. It is a prominent example of an intractable conflict with scarce intergroup communication that shares characteristics with other long-lasting conflicts around the world [25,26]. (The two groups in the conflict are commonly referred to as ‘Palestinians’ and “Israeli Jews”. The term ‘Israeli Jews’ is used to exclude Palestinians who have Israeli citizenship. Palestinians who remained within Israel’s borders in 1948 (after Israel’s war for independence and the Palestinian “Nakba” [27]) were granted citizenship in Israel.) Israeli Jews are in a state of protracted, violent conflict with Palestinians in the West Bank and Gaza. Palestinians in the West Bank live under military law and frequently experience negative contact with Israeli soldiers who guard checkpoints [28]; while Palestinians in Gaza live under blockade [29]. The belief in peace between the groups has been declining since 2016 [30], and the current war cycle that broke on October 7th, 2023 is thought to drive a resolution further away [31]. An indirect form of contact could be especially potent in such seemingly hopeless situations. However, it also requires careful planning and consideration due to its sensitive nature. We introduce two research questions:
**RQ1 (Acceptance):** How do Israeli Jews and Palestinians view the notion of telerobotic communication and its prospects as a medium for intergroup contact?**RQ2 (Preference):** Based on the preferences of Israeli Jews and Palestinians (how would their telerobot look like? What would it do?) derive design considerations for telerobotic-based intergroup contact?

From our conceptual model for telerobotic contact [24], we derive the following hypotheses:

**H1.** *General attitudes toward telerobotics influence the acceptance of intergroup telerobotic contact*.

**H2.** *General attitudes toward the outgroup influence the acceptance of intergroup telerobotic contact*.

Few existing studies have looked at attitudes toward robots among Israeli Jews and Palestinians, and previous research is inconclusive. For Israeli Jews, a study found positive attitudes toward robots [32]; the Technology Acceptance Model (TAM) predicts higher acceptance due to increased accessibility and previous exposure to novel technologies [33,34,35]. However, another study found native Israeli Jews to be among the least trusting of robot peacekeepers [36], and cultural values stemming from traditional Jewish law or Halacha, predict ethical concerns with artificial humanoids in the stories of Golem [37,38]. For Palestinians, Arabs from al-Sham countries (Jordan, Lebanon, Palestine, and Syria) were found to have a more positive attitude toward robots than Arabs from Africa, and positive acceptance of robots was found in Arab [39] and Muslim [40] cultures. Furthermore, studies have shown that disadvantaged groups ascribe a greater weight to the perceived usefulness, value, and well-being of technology when considering its acceptance [41,42,43], and that the effect of usefulness surpasses concrete access barriers [44,45], which could predict higher acceptance of communication robots. Without a clear prediction to one side or the other, we do not assume any difference between the populations in the result of RQ1.

A particular aspect to explore is the perceived intimacy of the telerobotic medium by Palestinians. Normalization of relations with Israeli Jews is often disapproved in Palestine [46,47]. Reports show that the more a medium is considered intimate, the less support it receives in society. In the qualitative portion of our survey, we try to assess how participants view telerobotic communication compared to other mediums. We define H3 as follows:

**H3.** 
*For Palestinians, the perception of telerobotic communication as a less intimate medium is associated with a higher likelihood of its acceptance for intergroup contact.*


Reports of organized interventions between Israeli Jews and Palestinians in the West Bank reveal that each group has its own needs and preferences for the type of contact [48,49]. According to the experimental research of [50], the advantaged Israeli group tends to focus on intergroup reconciliation, commonality, and acceptance, while the disadvantaged Palestinian group wishes to get empowered by acknowledging the asymmetrical power relations between the groups and confronting them directly [51]. An extended model [52] introduces a ‘dual’ form, in which the advantaged group also experiences victimhood and seeks to empower its sense of agency when confronting the outgroup. To evaluate these differences, we added open questions about how participants envision the telerobotic encounter. We define H4 as follows:

**H4.** *Different needs and preferences for intergroup contact of each group are reflected in how participants envision telerobotic intergroup contact*.

In our theoretical work [24], we looked at design factors that may be consequential for telerobotic intergroup contact. According to the social identity model of deindividuation effects (SIDE) [53,54], remote intergroup communication increases the likelihood that ingroup members adhere to their group identity. This has a positive side effect [55,56]: when group identity is salient during contact, participants are more likely to generalize positive attitudes and reduce prejudice toward the outgroup. Ingroup cues in the robot’s appearance, such as display of nationality or religion are therefore likely to be salient in remote communication as part of strategic self-expression, particularly in repressed groups [53,57,58,59]. We expect to see an increase in such cues in the intergroup scenario. We define H5 as follows:

**H5.** 
*The preference to display nationality and religion via the telerobot is greater in the intergroup scenario compared to the casual scenario.*


However, some traits have a reverse effect, as they are considered illegitimate stereotypes in the intergroup context. Although a nonhuman telerobot form may have a positive effect on the perceived senses of presence and self-extension in a casual conversation [60,61], in the intergroup context, viewing the outgroup as an animal or a mechanistic automaton is a common form of group dehumanization [23,62] and may invoke a negative connotation. Therefore, we hypothesize that participants would prefer the human form, when meeting the outgroup via a telerobot, to not appear dehumanized. We define H6 as follows:

**H6.** 
*Participants are less likely to prefer a nonhuman form for their telerobot in the intergroup contact scenario compared to the casual scenario.*


According to SIDE, anonymity in communication provides an additional layer of protection and freedom for participants to express their group identity [53,58]. In the survey, we asked participants to choose whether to show their face when operating the telerobot. We expect to see an increase in the preference for anonymity in the intergroup contact scenario. We define H7 as follows:

**H7.** 
*The preference for anonymity when operating a telerobot is greater in the intergroup scenario compared to the casual scenario.*


Anonymity is also a factor in the aforementioned illegitimate cues. Participants are more likely to portray such cues when they are anonymous [53,54]. Therefore, we hypothesize that telerobot operators who prefer not to disclose their identity are more likely to prefer a nonhuman form for their telerobot than those who are identifiable. We define H8 as follows:

**H8.** 
*The preference for nonhuman forms for the telerobot in the intergroup scenario increases when the users are not identifiable.*


Since being identifiable has the advantages of increased engagement and social presence [63,64], but may also prevent users from expressing themselves freely, we offered a third option in this study—portraying the telerobot as a puppet rather than as a representation of the teleoperator. This way, we allow operators to maintain (and optionally reveal) their original identity alongside the telerobotic puppet that has a separate identity. The use of such a hybrid identity in puppetry [65] supports the ‘dual identity’ strategy in intergroup contact [56]. In this strategy, ingroup members maintain the salience of their group identity, while also sharing a more common superordinate group with the outgroup (for example, being co-actors in a puppet play). The ‘dual identity’ strategy offers a balance between reconciliation and social action [66]. We assessed whether the puppet scenario would motivate more participants to join an intergroup conversation. We define H9 as follows:

**H9.** 
*The telerobotic puppet scenarios increase the acceptance of intergroup telerobotic contact.*


We analyzed the survey using a mixed-method approach, zooming in and out of the numerical data to link statistical trends to subjective content written in the open questions. The results of our study provide the first insight into the use of telerobotics for intergroup communication. We found that existing models of CMC and intergroup contact could be applied to telerobotic design and that they have particularities over traditional communication models. We also found significant differences between Israeli Jews and Palestinians in their perception of telerobotic communication and in their willingness to communicate with the outgroup.

## 2. Materials and Methods

### 2.1. Procedure

In March 2021, an online, anonymous survey was launched simultaneously in Israel (targeting only Jews) and Palestine (West Bank). The survey was open for one month. We reached out to participants in Israel through a surveying company and in Palestine through a paid research assistant. While the Israeli company maintains a database of a representative sample of the Jewish population, we were not able to obtain access to such a database in Palestine. Instead, we focused on convenience sampling, comprising a heterogeneous group of age and gender in both populations. Due to the lack of survey infrastructure, convenience sampling is often used in survey studies conducted in the West Bank [67,68,69]. The survey included binary/scale questions and open-ended questions. To combine the two types of questions into a unified theory, we used a mixed-method approach [70] to analyze the survey. To detect prominent themes in open-ended questions about attitudes toward telerobotics, we constructed a word cloud and compared the findings with quantitative data. Elsewhere, we were guided by the principle of ‘zooming in and out” [71,72]. In this process, we value every subjective response as a concrete projection of social phenomena. Therefore, we use individual data to look for patterns and use patterns to study individual data from a specific theoretical lens. Practically speaking, we analyzed open-ended responses from participants that stood out from the statistical data. We used qualitative findings to enrich the data supporting or opposing the hypotheses presented.

The survey was divided into four main parts: (1) demographic information, (2) general attitude toward robots and the outgroup, (3) opinions and preferences on casually meeting people via telerobotics, and (4) opinions and preferences on the use of telerobots for intergroup contact (i.e. the other group in conflict). The survey was conducted in Hebrew and Arabic and answers to all open questions were translated into English for analysis. The survey was anonymous, but participants were asked to specify their age, gender, spoken languages, perceived technological efficacy, previous robot experience, and previous intergroup communication experience. The survey allowed participants to skip questions, for example, if they expressed no interest in intergroup contact or telerobot operations, no further questions about their preferences regarding these issues were asked. Hence, the sample sizes vary for different questions. The survey was carried out during the ongoing struggle against the global COVID-19 pandemic. The COVID-19 pandemic put telerobotics (and remote presence in general) to center stage, as researchers sought solutions for remote operations in healthcare, work and education [73]. The survey also took place roughly one month before the tensions between Israeli Jews and Palestinians in Jerusalem reached another boiling point, leading to an extensive military conflict with Hamas in Gaza (dubbed Operation Guardian of the Walls), marking yet another escalation of violence [74]. Due to the sensitive nature of the survey, we took = measures to ensure that participants were comfortable with the questions and did not exit the survey out of frustration. This included skipping follow-up questions on scenarios that participants already specified that they were not interested in and removing some of the robot attitude scale questions to shorten the response time.

The general attitude toward robots was measured using an adapted NARS scale (Negative Attitude toward Robots Scale) [75] from the work of Tsui et al. [76] on attitudes toward telepresence robots. We included the S1 (interaction) and S3 (emotion) subscales, and neglected the S2 (social) scale for brevity. A higher ranking on the scale indicates a more negative attitude toward robots. The Cronbach alpha measures for the results were 0.8 for S1 and 0.86 for S3. Attitudes toward the outgroup were measured using a feeling thermometer. Feeling thermometers are a common measurement of explicit prejudice toward social groups [77,78,79]. Participants rate their attitude toward the group on a scale from zero (not favorable) to 100 (most favorable) (We decided to use the term ‘favoring’, rather than ‘warm’” or ‘cold’ because it translated more naturally to Hebrew and Arabic). Additionally, we asked participants to rate their attitude toward robots using a feeling thermometer. This method was previously used with robots by Crawford and Brandt [80].

To assess participants’ general interest in telerobotics, they were first presented with an illustration depicting the scenario of operating a telerobot situated in a remote location via a mobile phone (see Figure 1). Participants ranked their level of willingness to a number of telerobotic use cases, rated on a scale from 1 (*Definitely not*) to 5 (*Definitely yes*): (1) I would approach a telerobot at a public space, (2) I would use one in my home, or (3) I would be interested in operating a telerobot in a remote location. They also answered open questions about their perceived difference between a robot-mediated interaction and face-to-face interaction, what places would they like to visit with a telerobot, and how they would use it.

The next set of questions had to do with preferences on the appearance of the teleoperated robot. If participants expressed no interest in operating a telerobot (that is, if they indicated ‘*probably not*’ or ‘*definitely not*’), those questions were skipped. Participants were asked to choose one of the following appearance options: human, animal or plant, machine, or abstract shape. Following their selection, they were asked how their robotic avatar would reflect their identity. We offered multiple aspects in which the robot could appear similar to its operator: nationality, age, gender, voice, religion, or other physical accessories that reflect one’s identity. Participants were also able to select whether they would reveal anything about themselves or try to be someone else through the robot.

Next, we asked participants what features they would like to have in their telerobot. Participants rated a list of features on a scale from 1 (not at all important) to 5 (very important). The list included common social robot features [81] and features that we identified in our theoretical work [24] as those that may benefit intergroup contact scenarios. These included speech translation, moving around the space, making hand gestures, providing tactile feedback, showing emotions, showing online content (such as videos and images), showing personal content (such as a private photo file), playing games, and suggesting conversation topics. Additionally, we presented three options of anonymity during the conversation: only for the operator (the face of the operator is not visible, but they can see the interlocutor via the camera), for both (the face of the operator is embedded in the robot), or for none. The last question in this section allowed participants to type freely what other features they would like.

Once the participants completed the sections on telerobot and avatar design preferences, they were presented for the first time with the idea of using a telerobot to communicate with members of the outgroup. We first asked about the general attitude toward the outgroup using the feeling thermometer mentioned above and then about the level of interest on a scale from 1 (*Definitely not*) to 5 (*Definitely yes*) in meeting a member of the outgroup via four scenarios: a telerobot operated remotely via a smartphone, a robotic puppet performing remotely in front of an audience at a public location, a face-to-face meeting, and an online video chat. Participants who marked either *probably not* or *definitely not* on both of the telerobot-based forms skipped to the end of the survey. Those who expressed at least some interest in meeting an outgroup member using a telerobot were then presented with a similar set of questions regarding the preferred identity of their avatar and the preferred feature set of the telerobot when meeting the outgroup. Those interested in the puppeteering option were also presented with a symmetric scenario that included performing with telerobots in two locations simultaneously with a partner from the outgroup (see Figure 2). We also presented two additional questions. The first was to select the type of performance: political satire, children’s theater, adult drama, situational comedy, or other; the second was an open question asking participants to describe the desired type of performance in more detail. Finally, participants were presented with two open questions about their hopes and concerns when meeting a member of the outgroup through a robot.

### 2.2. Participants

A total of 617 participants completed at least part of the survey (321 in Israel and 296 in Palestine), of which 551 respondents completed the entire survey (286 in Israel and 265 in Palestine). According to the Palestine surveyor, a maximum of five participants may have originated from East Jerusalem rather than from the West Bank. Of the total 617 respondents, 304 were men (161 in Israel and 143 in Palestine), and 311 were women (158 in Israel and 153 in Palestine); two participants did not disclose their gender (both from Israel). The participants’ ages ranged between 15 and 71 years (M = 35.55, SD = 13.36); between 15 and 71 in Israel (M = 36.57, SD = 13.99); and between 15 and 71 in Palestine (M = 34.45, SD = 12.57). The Palestinian group was younger (t(615) = 1.9986, *p* = 0.0475) with no significant difference in genders (t(615) = 0.89, *p* = 0.374).

## 3. Results

In the results, the **IL** code denotes Israeli Jews and the **PS** code denotes Palestinians.

Due to the age difference between the groups, all ANOVA comparisons include age as a covariate.

### 3.1. Technological Self-Efficacy and Intergroup Contact Experiences

Israeli Jews (M = 3.8, SD = 1.05) reported a significantly higher level of technological self-efficacy compared to Palestinians (M = 2.88, SD = 1.33), F(1, 649) = 106.4, *p* < 0.001, η_p_^2^ = 0.14. This was consistent with Israeli Jew participants’ self-reported ownership of technical devices. Israeli Jews were significantly more likely to own a smartphone (F(1, 548) = 13.07, *p* < 0.001, η_p_^2^ = 0.023) or a computer (F(1, 548) = 168.96, *p* < 0.001, η_p_^2^ = 0.24) than Palestinian participants, and were more likely to have previous interactions with robots (F(1, 548) = 43.26, *p* < 0.001, η_p_^2^ = 0.07). Israeli Jew participants also had significantly more interaction with the outgroup than their Palestinian counterparts, either participating in organized encounters (F(1, 548) = 23.62, *p* < 0.001, η_p_^2^ = 0.04) or having friends from the outgroup (F(1, 548) = 16.63, *p* < 0.001, η_p_^2^ = 0.03).

### 3.2. General Attitude toward Robots and the Outgroup

#### 3.2.1. General Attitudes toward Robots

Palestinians had a considerably more favorable attitude toward robots than Israeli Jews, regardless of age or technological self-efficacy. This includes NARS scores (IL M = 0.4, SD = 7; PS M = −3.3, SD = 6.67, F(1, 613) = 96.12, *p* < 0.001, η_p_^2^ = 0.14; lower is more favorable), and the feeling thermometer (IL M = 56, SD = 25.1; PS M = 66, SD = 23.4, F(1, 608) = 62.32, *p* < 0.001, η_p_^2^ = 0.09; higher is more favorable). Comparisons include age and technological self-efficacy as covariates. There were no gender differences among Palestinians, but Israeli women (NARS M = 1.72, SD = 6.39; feeling thermometer M = 50.54, SD = 24.6) expressed a significantly more negative attitude toward robots than Israeli men (NARS M = −0.92, SD = 7.41; feeling thermometer M = 61.6, SD = 24.18), on both NARS: t(307.9) = 3.4, *p* < 0.001, and the feeling thermometer: t(312.6) = −4.02, *p* < 0.001.

#### 3.2.2. General Attitudes toward the Outgroup

Both groups indicated negative attitudes toward the other group, as indicated by low values on the feeling thermometer (overall M = 26.21, SD = 24.75; on a scale of 1 to 100). Nevertheless, the attitude of the Palestinian participants toward Israeli Jews (M = 12.92, SD = 17.28) was significantly lower than that of the Israeli Jew participants toward Palestinians (M = 38.7, SD = 24.22): F(1, 562) = 208.82 *p* < 0.001, η_p_^2^ = 0.271.

### 3.3. General Opinions on Meeting People via Telerobotics

#### 3.3.1. Interest in Telerobot Interaction and Operation

Interest scores for interaction and operation scenarios are displayed in Figure 3. Palestinians were significantly more interested in approaching a robot than Israeli Jews (F(1, 586) = 22.36, *p* < 0.001, η_p_^2^ = 0.037). The effect manifested primarily when comparing the women of both groups(1, 294) = 26.68, *p* < 0.001, η_p_^2^ = 0.083. The mean response from Palestinian women was in the moderate positive range (M = 3.55, SD = 1.13; 3 indicating *undecided*), while the mean response from Israeli women was slightly negative (M = 2.9, SD = 1.13). Age did not affect the responses of Palestinian participants, but older Israelis responded more positively than younger Israelis (r = 0.2, *p* < 0.001). Israeli Jews, however, were significantly more interested than Palestinians in operating a robot F(1, 586) = 7.95, *p* = 0.005, η_p_^2^ = 0.013. This effect manifested primarily among men F(1, 287) = 12.63, *p* < 0.001, η_p_^2^ = 0.042. Israeli men had a slightly positive mean view (M = 3.37, SD = 1.2; 3 being *undecided*) compared to the Palestinian men, who averaged a slightly negative view (M = 2.86, SD = 1.3). While age did not effect the responses of Israeli Jew participants, younger Palestinians responded more positively to the telerobot operation scenario than older Palestinians (r = −0.14, *p* = 0.02).

A more positive attitude toward robots on the NARS scale correlated with greater willingness to participate in all scenarios (Approaching: r = −0.57, *p* < 0.001; Having at home: r = −0.47, *p* < 0.001; Operating: r = −0.46, *p* < 0.001)). However, in some cases, even participants with a maximum favorable attitude toward robots (100/100 on the feeling thermometer) were deterred from an encounter via a telerobot. This was especially noticeable on the Palestinian side, with one participant noting that “*the idea of meeting [a] robot is brilliant, but the idea of it being controlled, I didn’t like it*”, and another noting that the “*aim of robots to make people’s lives easy, there are so many ways to use them other than in meetings*”.

#### 3.3.2. Comparing Telerobot Communication to a Face-to-Face Meeting

We asked participants how meeting a person via a telerobot would compare to meeting that person face-to-face. The most common word among Israeli Jew participants was ‘strange’ (n = 96/299: in 32% of the responses). For Palestinian participants, there was a more balanced response of positive and negative words. Figure 4 shows a comparison of word clouds. Of the 76 occurrences of ‘strange’ in Israeli Jewish participants, 19 had the form of ‘strange but ‘...’, adding something positive to the opinion. Of those, nine responses were in the form of *Strange at first but we could get used to it*, all were from Israeli women. Of the 14 occurrences of ‘strange’ in Palestinian participants, two had abated the claim with an additional positive comment. Overall, in response to RQ1, attitudes toward robots and telerobots differed greatly between Palestinians and Israeli Jews, with Israeli Jews having a more negative attitude toward robots and a more wary approach to telerobotics.

### 3.4. Interest in Meeting the Outgroup via Telerobotics

We asked participants if they would meet members of the outgroup through four different media: A telerobotic avatar (operating a telerobot at the location of the outgroup), a telerobotic puppet show, a face-to-face meeting, or a video call. The mean results of all participants in all scenarios were less than 3 (indicating *undecided*) and the results of the Israeli Jew participants were significantly higher than the mean results of the Palestinian participants (see Figure 5). Palestinians preferred to meet Israeli Jews via a telerobotic scenario compared to a video call or face-to-face (Wilcox signed rank V = 3320.5, *p* = 0.02), while Israeli Jews preferred the non-robotic scenarios (Wilcox signed rank V = 1774, *p* < 0.001). Both Israeli Jew and Palestinian participants significantly preferred the standard telerobotic scenario over the telerobotic puppet (telepuppet) performance (Wilcox signed rank IL: V = 1008, *p* < 0.001 PS: V = 580, *p* = 0.006).

A more positive attitude toward robots on the NARS scale did not predict the willingness of Palestinians to meet the outgroup via telerobotic scenarios, but did increase the willingness of Israeli Jews in both the telerobotic avatar scenario (r = −0.25, *p* < 0.001) and the telerobotic puppet theater scenario (r = −0.15, *p* = 0.01). Therefore, H1 is accepted for Israeli Jews and rejected for Palestinians. Furthermore, the willingness to operate a robot in a casual scenario also predicted the willingness of Israeli Jew participants to operate a telerobot in Palestine (r = 0.21, *p* < 0.001). Despite that, only within the Israeli Jew participants who selected the highest interest (5—*Definitely yes*) in casually operating a telerobot (N = 21/290: 7.2%), the interest in meeting the outgroup via telerobotics (M = 2.95, SD = 1.46) exceeded that of video (M = 2.8, SD = 1.47) and face-to-face (M = 2.76, SD = 1.4). Although there was no correlation between age and preference for interaction modes for Israeli Jew participants, younger Palestinians were more willing to meet Israeli Jews in the video call (r = −0.22, *p* < 0.001), face-to-face (r = −0.26, *p* < 0.001), and telerobotic puppet scenarios (r = −0.2, *p* = 0.001), but were not significantly correlated for the standard telerobotic scenario. Finally, a more favorable score toward the outgroup in the feeling thermometer increased the willingness of participants to converse with the outgroup on all media, including telerobotic avatars (IL: F(1, 288) = 94.5, *p* < 0.001, η_p_^2^ = 0.25; PS: F(1, 267) = 60.35, *p* < 0.001, η_p_^2^ = 0.18). Therefore, we accept H2.

### 3.5. Preferences for Telerobotics in Casual and Outgroup Communication

#### 3.5.1. Small Sample Size

Due to the large number of participants who were not interested in intergroup telerobotic contact, the number of participants who answered the preference questions that followed is substantially lower than the total number of participants. Of the total number of participants, 162 Israeli Jews and 28 Palestinians submitted their preference for telerobotic intergroup contact. Of these, 115 Israeli Jew participants and 21 Palestinian participants were also interested in the casual telerobotic operation, so their responses could be compared between the two scenarios. For this reason, we took special care that the results from the smaller Palestinian group meet strict standards. For the Wilcoxon signed-rank test, we assert that the test statistic value of the test (V) is less than or equal to the Wilcoxon signed-rank critical value corresponding to our sample size [82]. For the chi-square test, we ensure that none of the expected frequencies in our contingency table is less than 5.

#### 3.5.2. Telerobot Appearance

For a casual encounter, there was no statistically significant difference between the choices of the two groups. Human was the most favorable form, chosen by 52% of the Israeli Jewish participants and 42.75% of the Palestinian participants. ‘Machine’ was the second-best in both groups, chosen by 22.45% of Israeli Jew participants and 32.82% of Palestinian participants. For an encounter with the outgroup, Israeli Jew participants increased their inclination toward a human form to surpass that of Palestinian participants (χ^2^(1) = 16.28, *p* < 0.001, expected frequencies > 5, F(1, 187) = 17.82, *p* < 0.001, η_p_^2^ = 0.09), while Palestinian participants largely stuck with their original choice distribution. Therefore, we accept H6 only for Israeli Jews and reject it for Palestinians. Of the 27 Israeli Jew participants who chose a machine appearance for a casual telerobotic encounter, 11 (40%) changed their appearance to a human when meeting the outgroup. We evaluated whether the choice of visibility affected the choice of appearance for Israeli Jews in the outgroup scenario. Participants who opted not to be seen by the interlocutor had a significantly higher preference for a nonhuman appearance (χ^2^(1) = 8.04, *p* = 0.004). The effect was not present in the casual scenario; therefore, H8 is accepted for Israeli Jews.

#### 3.5.3. Identity Representation in the Telerobotic Avatar

When designing the identity of the telerobot, the desire to represent the nationality via the robot increased significantly increased in the outgroup scenario compared to the casual scenario (see Figure 6). This was true for Israeli Jew participants (Wilcox signed rank V = 126, *p* < 0.001) and Palestinian participants (Wilcox signed rank V = 36, *p* = 0.006, critical value for n = 21 and α <= 0.01: 42). Thus, H5 is accepted for both groups, but only for nationality and not for religion.

The outgroup scenario also significantly decreased the wish of both groups to be someone else through the robot (Israeli: Wilcox signed rank V = 286, *p* < 0.001; Palestinians: Wilcox signed rank V = 36, *p* = 0.006, critical value for n = 21 and α <= 0.01: 42).

#### 3.5.4. Visibility or Anonymity

We asked the participants if they would like to see and be seen by their interlocutor when operating the robot (through an embedded camera display), or whether their anonymity and/or their partner’s anonymity should be preserved. In both scenarios, participants preferred to see and be seen by their conversation partner (Casual: IL 76.3% N = 194, PS 76.4% N = 127; Outgroup: IL 76.2% N = 160, PS 46.4% N = 28). Comparing the answers to the casual and outgroup scenarios, we observe an increase in the number of choices to maintain anonymity on both sides when meeting the outgroup (IL from 1.7% to 8.8% N = 115; PS from 4.8% to 23.8% N = 21), but only the increase of Israeli Jew participants was statistically significant (Wilcox signed rank V = 0, *p* = 0.006). Thus, H7 is accepted only for Israeli Jews. A high percentage of participants who preferred to try a completely new identity or to not reveal their identity via the telerobot also chose to *be seen* by their interlocutors via the video feed. This was the case for 76.5% of Israeli Jew participants (49/64) and 73% of Palestinian participants (54/74) in the casual scenario, and 64% of Israeli Jew participants (23/36) and 33.3% of Palestinian participants (3/9) in the outgroup scenario.

#### 3.5.5. Features Preferences for the Telerobot

Participants freely ranked every proposed feature on a scale from 1 (not at all important) to 5 (very important) without any limit on the total pool of points. Therefore, to avoid aavoid bias caused by participants who consider any feature important, we chose to analyze each participant’s relative feature rank—the score of a feature relative to the mean score of all features. The “speech translation” feature was the most requested feature by both groups. This was followed by the ability to detect touch feedback and the ability to travel around the space. However, the relative ranking of the travel feature was significantly decreased for Israeli Jew participants in the outgroup scenario (Wilcox signed rank V = 3336, *p* < 0.001). On the contrary, the ranking of the “suggesting topics” feature for Israeli Jew participants increased in the outgroup scenario (Wilcox signed rank V = 1896, *p* = 0.043).

Participants were asked to speculate on what kind of abilities they would endow their telerobot with. Themes for the casual scenario were diverse in both groups, including abilities such as flying, cleaning, emotional expression, and heavy lifting. In the outgroup scenario, the answers consolidated to several main themes: abilities related to war, such as fighting, self-defense, and damage control; abilities related to translation and knowledge of the other’s culture and dialect; and abilities related to mediation, helping with the conversation, and representing the operator.

Responses for the outgroup scenario included varying degrees of optimism. Israeli Jew participants focused more on portraying good intentions, asking the telerobot to “show that we are good and we are all ultimately people”, to have a “sensation of goodwill” or that it could “make peace”. On the Palestinian side, responses were more critical, for example:

“*Ability to speak the Hebrew language and to have information about the Palestinian people and the suffering they endured from the Israelis*.”

#### 3.5.6. Hopes and Concerns

We asked participants to describe in two questions what their hopes and concerns would be about meeting the outgroup via a telerobot. Due to the generally lower interest of Palestinians in meeting Israeli Jews, only 27 Palestinian participants did not skip this part and answered both questions, compared to 159 Israeli Jew participants. Of the 25 Palestinians who did answer, only 6 used the ‘hopes’ question to disclose a positive message rather than express pessimism or reluctance about the proposed scenario. Hopes of Palestinians included “breaking mental borders”, “ability to communicate”, that “they [Israelis] can negotiate with him, not ignore him [the robot] or violate him”, and to “show that the Palestinians are resistant people”. Nearly a third of the Palestinian responses (8 out of 25) cited “normalization” as their main concern. The phrase “normalization is betrayal” was repeated twice (once with the hashtag sign: #التطبيع خيانة). We investigated the responses of these participants and found that seven out of eight participants who mentioned normalization had a negative (*probably not* or *definitely not*) response to meeting Israeli Jews via face-to-face or a video call, but *considered* meeting via a telerobot. Six participants chose *3-undecided*, *and one chose 1-definitely yes* for meeting via a telerobot. This participant (female, 42 years old) expressed the hope that the meeting would be less ’resonating [صداه]’ than a face-to-face meeting. This finding supports **H3**.

Hopeful responses from Israeli Jew participants contained general aspirations for peace, understanding, and reconciliation, but also specific references to how communication via telerobotics may aid in conflict resolution. Those included: Getting help from the telerobot in translation, mediation, and explanation; having less prejudice toward the robot than face-to-face; and the general possibility of having communication that otherwise would not have happened. One participant also noted that conversing in Palestinian territory may pose less risk of the meeting being intervened by the military; this participant had previously participated in organized intergroup meetings. Another participant hoped that “something in this encounter would be different from a physical encounter between people that usually lacks empathy.” The two most common concerns of Israeli Jew participants were about fear that the outgroup would destroy, damage, or hack the robot, or that the encounter with the telerobot would be superficial compared to a face-to-face meeting. More specifically, participants raised concerns over the lack of intimacy, lack of human gestures, that the use of a robot would appear cynical or patronizing, or that there would be too much focus on the technology rather than the humans behind it. One participant also expressed concern that the robot “would remind them [the Palestinians] of the weapons that are used against them.” On the contrary, some Israeli Jews noted that the gap created by the telerobot would make it easier for them to meet and relieve them of any fear.

#### 3.5.7. Interest in Telerobotic Puppetry

The interest of participants in both groups in performing a casual telerobotic puppet show was lower than the interest in other telerobotic scenarios such as approaching a telerobot, owning a telerobot, and operating a telerobot (combined mean = 2.37, SD = 1 compared to means of 3.22, 2.7, and 2.94), rejecting H9. However, 13 of 252 participants (5.1%) who expressed no interest (*probably not* or *definitely not*) in operating a telerobot chose *probably yes* for the casual telerobotic puppet show, and almost all (12) were Israeli Jews. The majority of those participants described meeting a telerobot as a strange, cold, and non-intimate encounter. The mean interest in the collaborative, symmetric puppet show with the outgroup was lower in both groups than the mean interest in the asymmetric telerobotic puppet scenario (IL 2.75 SD = 1.06 compared to 2.82 SD = 1.1; PS 2.3 SD = 1.07 compared to 2.74 SD = 0.98); however, the difference was not statistically significant.

When describing the collaborative puppet show, participants could select one or more genres they would be interested in performing. The top three choices were children’s theater, situational comedy, and political satire. Participants were also asked to describe in an open question what kind of show would be performed. Responses were classified according to the level of directness with which they refer to the Israeli–Palestinian conflict. Of the 38 responses from Israeli Jew participants, 26 (68.4%) did not refer directly to the conflict in their concept. Participants described reasons such as “I think I would like to do a show for kids that is not related to the conflict and is funny”, “Something that makes the kids laugh and would allow the performers to shake up the awkwardness”, and “characters of kids, that it [the story] would come from the most innocent and true place”. Of the two Palestinian participants who were interested and provided more details about the performance, one chose to avoid a reference to the conflict, describing ‘Characters of ordinary life”, while the other described a more direct and antagonistic concept:

*The story is about a religious idea and how they occupied our land by force, robbed our religious places, and many beliefs*. This participant (female, 26 years old) was not interested in operating a robot in a casual scenario and marked *0 out of 100* on the feeling thermometer toward Israeli Jews. For meeting Israeli Jews, she was undecided about other options, but marked *Definitely yes* for cooperating with an Israeli Jew in a joint telerobotic puppet show, thus supporting H9. Overall, the responses to the open questions on hopes, concerns and topics for puppetry support H4 on differences in preferences between the advantaged and disadvantaged groups.

## 4. Discussion

The results indicate substantial differences between the groups in the acceptance and attitudes toward robots in general, and in a willingness to meet the outgroup. Israeli Jews prefer more traditional communication media, such as video call or face-to-face; while Palestinians are more open to telerobotics. Yet, Palestinians express an overall lack of willingness to meet with the outgroup in any medium. The deterrence from telerobotics within Israeli Jews appears to stem from and be predicted by a generally negative attitude toward robots. Israeli Jew participants almost collectively described a casual interaction via telerobotics as “strange” and showed a lack of interest in interacting with a remotely controlled robot, especially among younger Israeli women. The gender difference is consistent with the results of Tsui et al. [76]. Israeli Jews also voiced concerns about the lack of intimacy in telerobotics for meeting the outgroup. At the same time, Israeli men showed positive interest in operating a telerobot in a casual scenario, but this preference did not translate to the outgroup scenario when compared to video and face-to-face, except for a small number of telerobotic enthusiasts. This may indicate that, although some Israeli Jew participants liked to experiment with telerobotics and were interested in meeting Palestinians in other media, they did not see telerobotics as an adequate or serious medium for intergroup contact.

The favorable attitude of Palestinians toward telerobotics supports our hypothesis on the perceived usefulness of technology within disadvantaged groups and the positive inclination of Arab and Muslim cultures toward robots. This study also revealed that participants did not want to communicate with Israeli Jews due to it manifesting as an unwanted normalization of relations, but found it easier and less resonating to communicate via telerobotics. Taking into account the findings of Mi’Ari [47], that support for normalization increases in fields where there is less intimacy involved, this may indicate that Palestinian participants perceive telerobotic communication to be less intimate than a video call and a face-to-face conversation. Unlike Israeli Jews, the Palestinian interest in intergroup telerobotics was not correlated with a general attitude toward robots. This may indicate that the lack of intimacy is a stronger motive for outgroup communication than the general interest in telerobotics. Nevertheless, the majority of Palestinians still leaned toward avoiding contact altogether, and were only slightly more interested in telerobotics than other proposed means. Communication interest in all mediums was strongly predicted for both groups by the general attitude toward the outgroup. This suggests that a telerobotic encounter should be carefully designed to target the right audience and to fit the needs of both parties to have the potential for reconciliation. Differences in responses to open questions about the robot’s features and participants’ hopes and concerns for interaction support previous research on the difference in needs between groups in conflict [48,49]. When designing the robotic encounter, Palestinians were generally more interested in confrontation and direct references to the conflict; while Israeli Jews hoped to experience commonality and elicit understanding via the robot.

When choosing an appearance for their telerobot in both the casual and outgroup scenarios, the human form was the most desirable form in both groups. Crucially, Israeli Jew participants who chose a nonhuman form in the casual scenario tended to change their appearance to a human when faced with the outgroup scenario, especially if their first choice had a machine-like appearance. This supports the findings of Shnabel and Nadler [49] and our hypothesis that a nonhuman form may resonate with the dehumanizing aspect of the conflict. In this case, the perpetrator group had a need to restore its moral image—to appear human. A similar phenomenon may have also occurred in the choice of robot features. Israeli Jew participants significantly decreased their preference for traveling around the space in the outgroup scenario compared to the casual scenario, perhaps refraining from demonstrating their mobility in Palestinian territory. For Palestinian participants, we did not see such inhibitions, which signifies a more assertive use of the medium. We also found support for models of group identity under anonymity [53,54] when Israeli Jew participants, who opted to not reveal their faces via the robot, were more likely to choose a nonhuman appearance in the outgroup scenario, possibly having fewer moral concerns under anonymity. Israeli Jews also increased their desire for anonymity when meeting the outgroup (although the preference to be identifiable was still the most popular in both groups). Both groups increased their desire to represent their nationality when meeting the outgroup via telerobotics, confirming the theory of increased group identity in mediated communication [58]. We observe that not only do Palestinians see their national identity as a means of empowerment, but so do Israeli Jew participants. This corresponds to the ‘dual’ model of Nadler and Shanbel [52], in which both the advantaged and the disadvantaged groups feel a threat to their agency and identity and wish to restore it through contact. Religion, however, did not play a strong role in the desire to portray an identity to the outgroup.

We found that most of those who opted to *not reveal* their identity through the robot’s appearance still chose to *be seen* by the interlocutor through the video feed. This could point out the complexity of wanting to be visibly present when interacting with the outgroup, yet not completely exposed. Due to its hybrid form, telerobotic puppetry may provide an answer to this need to separate the identity of the robot from that of the operator, but this potential did not manifest broadly in the survey results. The forms of telerobotic puppetry were less accepted than the more standard forms of telerobotics. However, the fact that participants opted to play with their identity via their robot warrants further exploration of this form and its potential for the ‘dual identity’ strategy [56]. Additionally, some Israeli Jews who found the idea of telerobotics strange and alienating were willing to try telerobotic puppetry. For one Palestinian participant, the idea of being able to tell the story of the Palestinian people in a performance converted her view from a complete refusal of any interaction to an agreement to collaborate on a performance together with an Israeli Jew. This study provides a first insight into the use of telerobotics for intergroup contact in the Israeli–Palestinian conflict and beyond. Although our results are in line with previous research and hypotheses, this study is not without limitations. First, the online survey was randomly distributed while controlling equal groups of age and gender, but did not use a representative sample of the Palestinian population. Therefore, it may exhibit some skew in the results and cannot be generalized to all Palestinians. The use of the Internet as a medium also immediately introduces bias regarding technological self-efficacy. Future studies could reify the results by using a representative sample. Second, as mentioned above, it was performed in a particular sociopolitical context (pre-war, amid COVID-19) which may have influenced the results. Finally, because only a fraction of the participants (especially on the Palestinian side) continued to the section of the survey that explores intergroup contact, the results in that section are even more vulnerable to statistical variance. Maintaining the balance between respecting the preferences of survey participants and obtaining enough data is challenging, especially in the sensitive context of intergroup conflict. In future studies, we could also collect data for those who are not interested in contact by asking them the reason behind their refusal. For this, we could use open questions that allow participants to express their opinion more freely [83]. Nonetheless, the results of this study make it clear that telerobotic communication technology has the potential to attract a wide audience and provide meaningful encounters in areas of conflict. We hope for robot designers, engineers, social psychologists, and peace-building practitioners to further develop this field.

## Figures and Tables

**Figure 1 behavsci-14-00854-f001:**
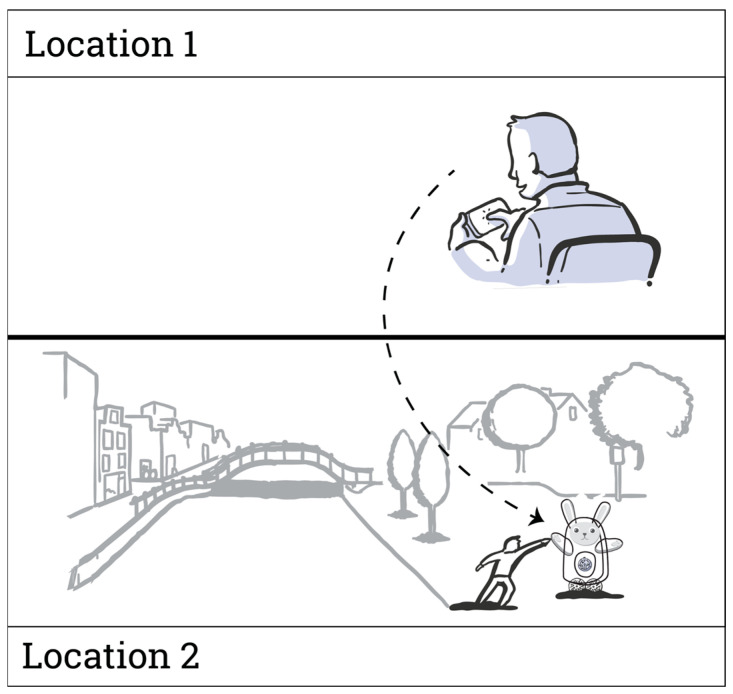
Survey illustration depicting a telerobotic scenario. A person in Location 1 is using their mobile device to operate a telerobot situated in Location 2.

**Figure 2 behavsci-14-00854-f002:**
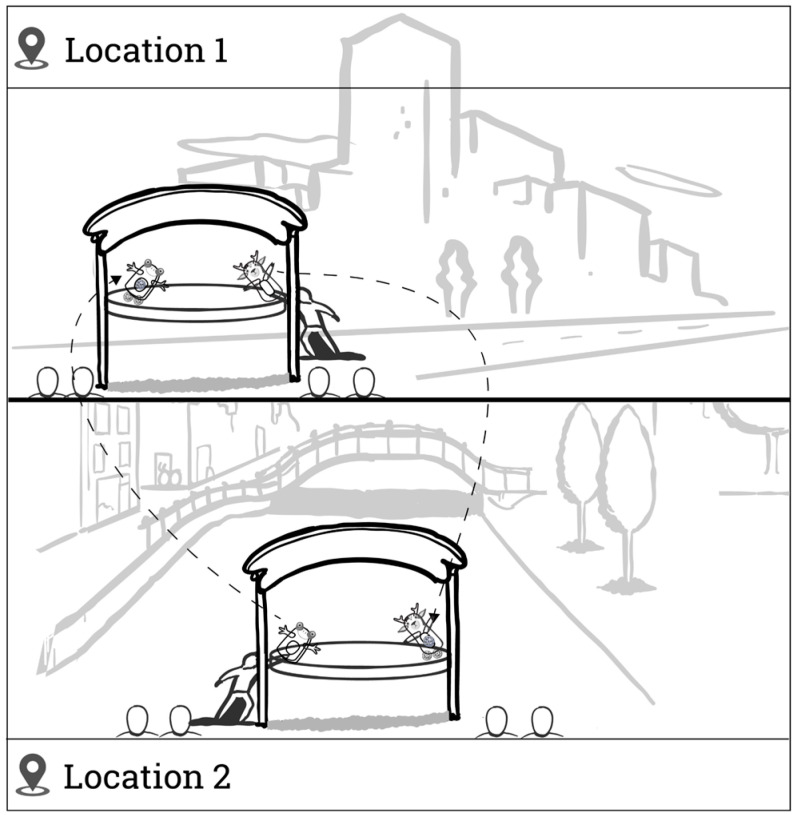
Survey illustrations depicting the symmetric collaborative telerobotic puppet scenarios. Puppeteers at Location 1 operate a puppet remotely at Location 2 and vice versa.

**Figure 3 behavsci-14-00854-f003:**
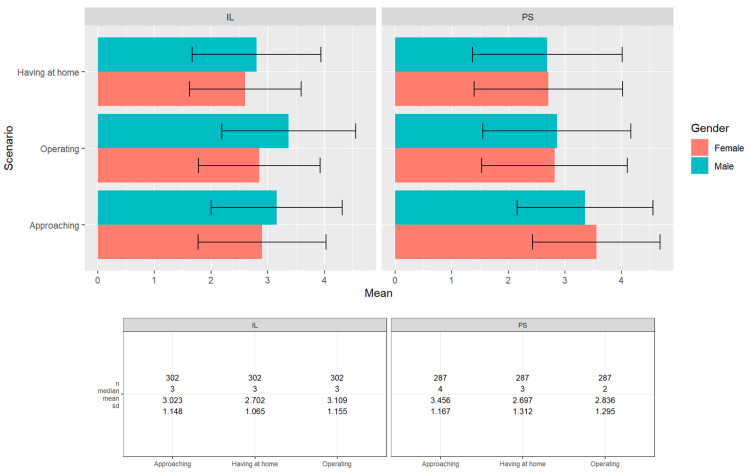
Interest in casual telerobotic scenarios (IL = Israeli Jews; PS = Palestinians).

**Figure 4 behavsci-14-00854-f004:**
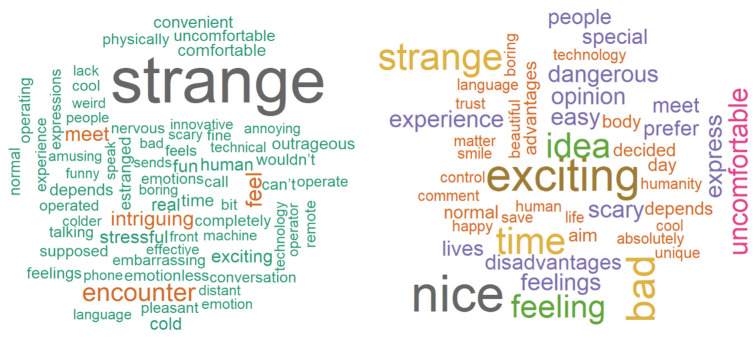
Comparison of telerobot-based communication to meeting face-to-face. (**Left**): IL; (**right**): PS.

**Figure 5 behavsci-14-00854-f005:**
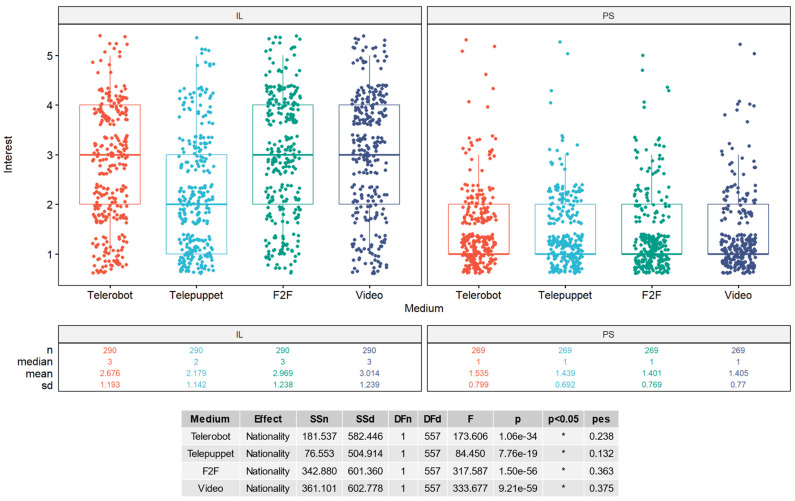
Interest in meeting the outgroup via different mediums.

**Figure 6 behavsci-14-00854-f006:**
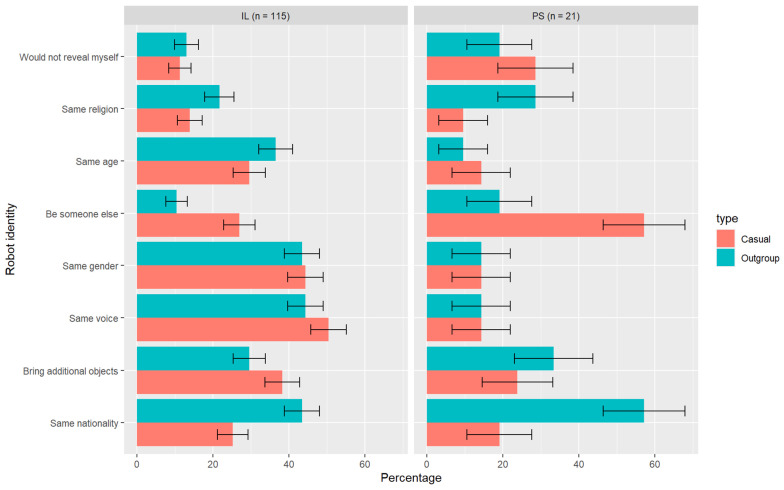
Identity representation in the casual and outgroup scenarios.

## Data Availability

Anonymous data and code for statistical analysis in the R software (v4.3.3) is available at: https://zenodo.org/records/11200209 (accessed on 14 May 2024).

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
