# Peer review of "Telerobotic Intergroup Contact: Acceptance and Preferences in Israel and Palestine"

_behavsci, 2024, doi:10.3390/bs14090854_

Round 1

Reviewer 1 Report

Comments and Suggestions for Authors

Review of Behavioral Sciences behavsci-3027160: Telerobotic intergroup contact

The paper presents results of a cross-sectional interview study based on 321 Israeli and 296 Palestine adult respondent about their attitudes towards telepresence robots, attitudes about the respective outgroups, preferred robots’ appearance, anonymity during the conversion and the respondents’ opinions and preferences for the use of telerobots for intergroup contact.

I cannot recommend publication of the paper in its present form. My main argument for this recommendation is that I could not understand what telerobot exactly looks like, what the robots are intended to do and how it will be tied to intergroup contact. In addition, the authors speak about an operator and an interlocutor (lines 50 ff.) without explaining what that means. In other words, the object of discussion is insufficiently explained.

A second general argument against publication of the paper in its present form is that the authors do not say anything about the representativity of the two samples that the authors use and compare with each other. Nevertheless, the authors report their results (lines 255 ff.; line 522 ff.) as if they were representative for the two populations of Jewish Israelis and Palestine people. This would prerequisite a demonstration of near-representativity.

In addition, I have some open questions and minor recommendations for improvement:

Lines 300 ff: Why where the valences of the different variable differently sored? For attitude towards robots a higher score indicates a more negative evaluation, for attitudes towards the outgroup a more positive one. This makes the data report difficult to understand.

How did the authors handle individual missing data (e.g. lines 286 ff.)? Different degrees of freedom indicate that there are a few.

Lines 315: The meaning of the qualitative data remains unclear for me.

Line 526: The authors speak about mediation processes. I didn’t find a mediation analysis in the result section.

Author Response

Thank you very much for your comments.

We hope that this revision is much clearer and easy to evaluate.

Below are specific responses to your comments:

Comment 1: I could not understand what telerobot exactly looks like, what the robots are intended to do and how it will be tied to intergroup contact. The authors speak about an operator and an interlocutor (lines 50 ff.) without explaining what that means. In other words, the object of discussion is insufficiently explained.

Response: We rephrased the introduction text to more clearly depict the surveyed scenario of telerobotic intergroup contact (paragraph starts on line 57). We also rephrased the research questions (L87) to clarify that the robot is tied to intergroup contact by acting is a medium for intergroup contact, and that the questions “what the telerobot looks like” and “what it is intended to do” are exactly the questions that we ask the participants.

Comment 2: The authors do not say anything about the representativity of the two samples that the authors use and compare with each other. Nevertheless, the authors report their results (lines 255 ff.; line 522 ff.) as if they were representative for the two populations of Jewish Israelis and Palestine people. This would prerequisite a demonstration of near-representativity.

Response : We added a remark about representativity in the Method section (L226). We address this as a limitation in the Discussion section (L739).

Comment 3: Lines 300 ff: Why where the valences of the different variable differently sored? For attitude towards robots a higher score indicates a more negative evaluation, for attitudes towards the outgroup a more positive one. This makes the data report difficult to understand.

Response: The NARS scale, that was used for studying attitude toward robots,  measures “negative access”, and is therefore opposite of the feeling thermometer which measures positive feelings. We rephrased the references to NARS (L281, L379, L416) to avoid this confusion.

Comment 4: How did the authors handle individual missing data (e.g. lines 286 ff.)? Different degrees of freedom indicate that there are a few

Response: The second measure here refers to comparing only women of both groups, explaining the different degrees of freedom. We rephrased the sentence to clarify this (L403).

Comment 5: Lines 315: The meaning of the qualitative data remains unclear for me.

Response: The sentence was rephrased to clarify the question that was asked in the survey (L432).

Comment 6: Line 526: The authors speak about mediation processes. I didn’t find a mediation analysis in the result section.

Response We changed the word “mediated” to “stem from and be predicted by”, to not create a false impression that we performed mediation analysis (L660).         

Reviewer 2 Report

Comments and Suggestions for Authors

Dear all,

I particularly appreciate your efforts in drafting this manuscript. I have therefore carefully reviewed this paper and would like to offer my feedback and comments.

The manuscript aims to explore the use of telerobotic communication to foster intergroup contact in a conflict situation between Palestinians and Jewish Israelis. Specifically, this manuscript investigates the acceptance and preferences of telerobotic communication between the two different groups.

The aim of the research thus seems clear. The application of telerobotics to inter-group contact is an innovative approach that takes profit from recent technological advances. The article places it in the broader context of the Israeli-Palestinian conflict, offering a new perspective on how technology can bridge divisions. Furthermore, using a mixed-methods approach, the Study combines quantitative and qualitative data, enhancing the depth and richness of the clearly presented analysis. The examination of various aspects of design, such as human or non-human form and identifiability, is well conceived and relevant. Finally, the conclusions are supported by the literature presented in the manuscript.

Given these strengths and the positive assessment of the work, I would only suggest three minor form changes:

1. I recommend rechecking throughout the text the punctuation marks used and topographical symbols (e.g.: some brackets are missing at the end)

2. For higher clarity and order, I would recommend better identifying the various hypotheses (exactly as was done for the two research questions)

3 Still for the purpose of higher clarity, I would recommend dividing the paragraph "2. Materials and Methods" into two sub-paragraphs identified as "Participants" and "Procedure".

In conclusion, the manuscript as a whole is a valuable contribution to the field. I hope that the adoption of the recommended changes will be helpful in enriching this work.

Author Response

Thank you very much for your comments.

We hope that this revision is much clearer addresses your concerns.

Below are specific responses to your comments:

Comment 1: I recommend rechecking throughout the text the punctuation marks used and topographical symbols (e.g.: some brackets are missing at the end)

Response: We ran another check for punctuation and grammar throughout the whole text.

Comment 2: For higher clarity and order, I would recommend better identifying the various hypotheses (exactly as was done for the two research questions))

Response: We broke out the research questions into ten hypotheses and answered them explicitly throughout the paper (Introduction section, Results section).

Comment 3: Still for the purpose of higher clarity, I would recommend dividing the paragraph "2. Materials and Methods" into two sub-paragraphs identified as "Participants" and "Procedure"

Response: We reorganized the structure as suggested.

Reviewer 3 Report

Comments and Suggestions for Authors

This was an interesting article to read, and I appreciated the chance to review it. I think there are some very cool findings that line up nicely with the needs of dominant vs subordinate groups (e.g. preferences for social presence, anonymity/representation/salience, suggesting topics), but I would like to see more of an effort to tie to that to specific findings or trends in intergroup contact research rather than the more casual allusions currently presented. I would suggest the salience piece is especially critical given that research suggests that intergroup contact cannot work without it, and the authors seem to suggest going with a route that would reduce salience (the puppetry option). It is, however, very interesting how people want to (unproductively in cases) play with salience and minimize group elements.

This study is very exploratory, which is okay, but in some cases the authors suggest that there are specific hypotheses to test (many coming from a past theoretical paper), and then don’t always provide clear tests of them. If the authors wish to be purely exploratory than they need to find clearer ways to articulate research questions, and the need for them (e.g. past research is mixed, or there is not theory, etc.). In terms of method, this is a single cross sectional study that has made the mistake (I would argue) or reducing their n (especially among Palestinians) through an unnecessary screening question. Just because they don’t want something doesn’t mean if forced they wouldn’t have preferences. As a result, of the very asymmetrical n I am very wary of overinterpreting Israeli/Palestinian comparisons. 

The addition of the qualitative data seems incredibly minimal and seems to lack rigor. I am not a qualitative scholar, but from my perspective framing this as mixed methods goes a little far. There are only occasional qualitative blurbs with no discussion of the specific qualitative analytical techniques used to pick them or highlight their significance. Is another reviewer is an expert on this in and out method I will, of course, defer to them. Still, I am generally okay with this if an exploratory study is something the editor wishes to publish. 

Below, I have some more specific comments, many of which relate to the above:

I have a general preference for when people cite primary sources rather than reviews, especially when there are specific empirical claims being made. I would encourage the authors to change some of their review citations (e.g.s 2, 3, etc.) with that in mind

Does citation 24 actually cover empirical evidence that contact has gone down significantly for the average Israeli or Palestinian? Honest question since I don’t have access to that text. It seems more like a political than empirical piece. Regardless, I am not totally sure how it impacts this piece as the data was collected before the conflict escalated.

L79: If the authors’ theoretical framework make a direct prediction, I would encourage them to re-summarize the argument here (if there is more to add, ideally in regards to social technologies rather than technology in general), and make it into an explicitly tested hypothesis in the context of this research.

L96: Again, I would write this as a more formal hypothesis (e.g. that both technological bias and outgroup bias will both negatively predict attitudes towards telerobotic contact). However, I am not sure of the theoretical importance of this hypothesis, so maybe the authors can expand on that. The following conversation that describes an interaction beween perceived affordances and group on acceptance really seems to be its own argument (and again should be formalized as a hypothesis).

L108: It is not clear to me what connection the authors are making between the content of the contact will relate to their feelings about the robotic elements of contact (but rather reflect more general intergroup preferences), especially given that they don’t measure content. However, there are differences observed from group to group that might be explained by general tendencies. However, I think to better explain that the authors should look into the literature on power and intergroup contact.

L118: The issues surrounding anthropomorphism and identity salience are really interesting. They should be explicitly linked to the literature on intergroup contact that discusses the importance of identity salience though.

L125: When the authors say that they expect the display of ingroup cues to be moderated by the perceived identifiability what do they mean? It makes more sense as predicted instead of moderated, but that is just a tautology anyway so I am a little lost. Similarly the next sentence doesn’t make sense to me, as I am not sure why their comfort with identification would have anything to do with how cues are interpreted by the interlocutor. I imagine this is tied to your notion that individuals might have a choice in how much is presented, but on the whole this whole argument is a little muddled and needs refinement. 

Given the Palestinians were slightly younger (L182), and age is correlated with technology acceptance, but also less likely to own technological devices and report self-efficacy (L255) the authors need to use age as a covariate in any group comparisons. They say they control for this in attitudes towards robots but they don’t report the ANCOVA. Given that age, at least, is a significant covariate in some analyses (e.g. L290) the ANCOVA should really be the default reported. Also, effect sizes should be reported for all statistics.

L186: The NARS scale has sub-dimensions. I am curious why the authors didn’t use them.

L192: Not favorable to most favorable is not the response scale for feeling thermometers. As a result, this represents a constructed variable by the authors rather than a previously validated one and that needs to be noted.

The authors need to report general (rather than solely group level) descriptive statistics of their measures (including reliability for multi-item scales).

L204: I don’t understand why the authors would screen participants out (based on definitely and PROBABLY not answers) before asking for ideal appearance. There may be important variance there they no longer have access to. How many were screened out? What was the rationale? The authors should discuss this as a potential limitation. I have similar concerns for why the screened out participants prior to asking about features. This creates major power issues and reduces the ability of the authors to offer a comprehensive theoretical picture (e.g. L360).

Figure 2. maybe this is just me, but the illustration seems to indicate two potential confounds. First, it looks aggressive. Were perceptions of the illustration piloted? Second there are people watching. This would have HUGE connotations in a conflict like the Israeli-Palestinian one. Was this considered at all? If not limitations definitely need to be discussed here.

Comments on the Quality of English Language

Well done.

Author Response

Thank you very much for your comments and in-depth review of our paper.

We hope that this revision is much clearer and that it addresses all of your comments.

Below are specific responses to your comments:

Comment 1: I would like to see more of an effort to tie to that to specific findings or trends in intergroup contact research. The salience piece is especially critical. The authors seem to suggest going with a route that would reduce salience (the puppetry option).

Response: You are right to interpret our analysis as supporting the route of telerobotic puppetry and it is indeed a topic we have explored and will soon be published in a new paper. We now connected the concepts of puppetry and group salience into intergroup contact research in the paragraph starting on L197.

Comment 2: In some cases the authors suggest that there are specific hypotheses to test (many coming from a past theoretical paper), and then don’t always provide clear tests of them. If the authors wish to be purely exploratory than they need to find clearer ways to articulate research questions.

Response: We broke out the research questions into ten hypotheses and answered them explicitly throughout the paper (Introduction section, Results section).

Comment 3: A mistake (I would argue) of reducing their n (especially among Palestinians) through an unnecessary screening question. Just because they don’t want something doesn’t mean if forced they wouldn’t have preferences. As a result, of the very asymmetrical n I am very wary of overinterpreting Israeli/Palestinian comparisons.

L204: I don’t understand why the authors would screen participants out (based on definitely and PROBABLY not answers) before asking for ideal appearance. There may be important variance there they no longer have access to. How many were screened out? What was the rationale? The authors should discuss this as a potential limitation. I have similar concerns for why the screened out participants prior to asking about features. This creates major power issues and reduces the ability of the authors to offer a comprehensive theoretical picture (e.g. L360).

Response: We added a remark describing the rationale behind our decision (L259): “Due to the sensitive nature of the survey, we took extra measures to ensure that participants were comfortable with the questions and did not exit the survey out of frustration. This included skipping follow-up questions on scenarios that participants already specified that they were not interested in and removing some of the robot attitude scale questions to shorten the response time.”

 We understand your concern, and this was admittedly a risk that we took for the sake of eventually getting more meaningful responses. We believe that our results are still significant and meaningful despite this limitation.

Comment 4: The addition of the qualitative data seems incredibly minimal and seems to lack rigor. I am not a qualitative scholar, but from my perspective framing this as mixed methods goes a little far. There are only occasional qualitative blurbs with no discussion of the specific qualitative analytical techniques used to pick them or highlight their significance

Response: We believe that now when the hypotheses are clearly defined, it is easier to see the contribution of the qualitative data. We also elaborated on our approach in the “Methods” section (L231-240).

Comment 5: I have a general preference for when people cite primary sources rather than reviews, especially when there are specific empirical claims being made. I would encourage the authors to change some of their review citations (e.g.s 2, 3, etc.) with that in mind.

Response: We changed that paragraph to include primary sources with studies that demonstrate the claims (L35-41).

Comment 6: Does citation 24 actually cover empirical evidence that contact has gone down significantly for the average Israeli or Palestinian? Honest question since I don’t have access to that text.

Response: Citation 24 was in fact an opinion article meant to predict the still unclear results of the current war. We now added an empirical reference about the declining belief in peace, and clarified that we do not make any factual claims about the rate of contact (L79-82).

Comment 7: L79: If the authors’ theoretical framework make a direct prediction, I would encourage them to re-summarize the argument here (if there is more to add, ideally in regards to social technologies rather than technology in general), and make it into an explicitly tested hypothesis in the context of this research.

Response: The argument was now made explicit via the hypotheses H1 and H2.

Comment 8: L96: Again, I would write this as a more formal hypothesis (e.g. that both technological bias and outgroup bias will both negatively predict attitudes towards telerobotic contact). However, I am not sure of the theoretical importance of this hypothesis, so maybe the authors can expand on that. The following conversation that describes an interaction beween perceived affordances and group on acceptance really seems to be its own argument (and again should be formalized as a hypothesis).

Response: We removed this hypothesis since it was made redundant in the new structure.

Comment 9: L108: It is not clear to me what connection the authors are making between the content of the contact will relate to their feelings about the robotic elements of contact (but rather reflect more general intergroup preferences), especially given that they don’t measure content. However, there are differences observed from group to group that might be explained by general tendencies. However, I think to better explain that the authors should look into the literature on power and intergroup contact.

Response: We clarified that the answer to this question is given by the open questions (hopes, concerns, puppet theater content) and added another reference about the role of power in intergroup contact (Paragraph on L137).

Comment 10: L118: The issues surrounding anthropomorphism and identity salience are really interesting. They should be explicitly linked to the literature on intergroup contact that discusses the importance of identity salience though.

Response: This clause referred to the issue with a humanoid robot’s identity overriding the robot operator’s identity in telerobotic communication - not in relation to group identity salience. The idea we try to convey is about the risk of dehumanization in nonhuman forms despite the advantages they may have in causal telepresence scenarios. We rephrased the sentence to make this clearer (Paragraph on L153).

Comment 11: L125: When the authors say that they expect the display of ingroup cues to be moderated by the perceived identifiability what do they mean? It makes more sense as predicted instead of moderated, but that is just a tautology anyway so I am a little lost. Similarly the next sentence doesn’t make sense to me, as I am not sure why their comfort with identification would have anything to do with how cues are interpreted by the interlocutor. I imagine this is tied to your notion that individuals might have a choice in how much is presented, but on the whole this whole argument is a little muddled and needs refinement.

Response: We refined the argument and presented it as H7, explicitly testing if the use of nonhuman (dehumanized) forms increases when the users are anonymous.

Comment 12: Given the Palestinians were slightly younger (L182), and age is correlated with technology acceptance, but also less likely to own technological devices and report self-efficacy (L255) the authors need to use age as a covariate in any group comparisons. They say they control for this in attitudes towards robots but they don’t report the ANCOVA. Given that age, at least, is a significant covariate in some analyses (e.g. L290) the ANCOVA should really be the default reported. Also, effect sizes should be reported for all statistics.

Response: We added age as a covariate and reported effect size in all group comparisons (staring L361). Results remained unchanged.

Comment 13: L186: The NARS scale has sub-dimensions. I am curious why the authors didn’t use them.

Response: We clarified that we used two out of the three subscales (L282). Looking at the combined result felt adequate for this purpose of general attitude toward robots and telerobotics. 

Comment 14: L192: Not favorable to most favorable is not the response scale for feeling thermometers. As a result, this represents a constructed variable by the authors rather than a previously validated one and that needs to be noted.

Response: We added a footnote (L288) explaining that we decided to use the term “favoring”, rather than "warm" or "cold " because it translated more naturally to Hebrew and Arabic.

Comment 15: The authors need to report general (rather than solely group level) descriptive statistics of their measures (including reliability for multi-item scales).

Response: We posted general Cronbach Alpha measures for the NARS scale (L284).

Comment 16: maybe this is just me, but the illustration seems to indicate two potential confounds. First, it looks aggressive. Were perceptions of the illustration piloted? Second there are people watching. This would have HUGE connotations in a conflict like the Israeli-Palestinian one. Was this considered at all? If not limitations definitely need to be discussed here.

Response: We ran the illustration by the local staff and translators and did not receive any remarks that it is aggressive. The idea of people watching is intended in the puppet theater scenario. The participants were presented with the concept of performing as a puppet theater publicly in front of an audience. We added a clarification in the text (L332).

Round 2

Reviewer 1 Report

Comments and Suggestions for Authors

I still don't understand the exact procedure of realizing telerobotic intergroup contact. 

At least the Palestine sample is not representative. Therefore all declarations about mean scores of the Palestine sample and comparisons between the samples have to be confined to the concrete Palestine sample and cannot be generalized over the whole Palestine population.

Author Response

Thank you for second review!

Below is our response to your comments:

Comment 1:

I still don't understand the exact procedure of realizing telerobotic intergroup contact.

Response 1:

We added on line 53 a description of how telerobotic intergroup contact might be realized. We hope that this clears up the confusion about this form of contact.

Comment 2:

At least the Palestine sample is not representative. Therefore all declarations about mean scores of the Palestine sample and comparisons between the samples have to be confined to the concrete Palestine sample and cannot be generalized over the whole Palestine population.

Response 2:

On line 254 we elaborated on the reasons behind the use of a convenience sample rather than a representative sample. We also included three new references to demonstrate that the use of convenience sampling is common for surveys done in the West Bank.

In the limitations section starting on line 734 we made the explicit statement that the the results cannot be generalized to all Palestinians.

Reviewer 3 Report

Comments and Suggestions for Authors

I think the authors have done a good job addressing a number of my concerns. I am still concerned about the screening criteria and assymetrical sample, but agree with the authors that the findings are still a contribution in spite of this (although I would still like more discussion of why this limitation matters). I still don’t think the qualitative elements seems rigorous enough to address the hypotheses. There is just a lack of text and methodological discussion here. I don’t think that should keep this from being published, but I would advise framing and discussing it as exploratory or taking it out. I appreciated the hypotheses being added, but in so doing it is clear that some of the arguments are still not fully articulated (but I think they could be).

H3 is a null hypothesis rather than a research hypothesis. Since the null hypothesis should not be interpreted I would suggest instead framing it as a RQ and not overinterpreting any lack of difference.

H4: The writing of this is confusing to me. Additionally, there is an extra colon.

H6 should be framed in terms of preferences rather than use (unless I am misunderstanding something). I also don’t know why the hypothesis is the way it is given a general propensity to want to dehumanize outgroups (especially when in heightened states of conflict).

Leading into H7, the group identity models should be unpacked if the authors wish to use them as the basis for a prediction. As is, I am not even really sure what is being said here. Perhaps this is in relation to reduced salience as well, but if so why is is different than H8.

In regards to discussing H8 the relation of salience to prejudice reduction should probably be discussed either here or in the discussion.

***As a note here the disconnect between H7 and H8, i.e. between interpersonal and group level anonymity is really interesting and it would be cool if the discussion leading up to H7,8,9 reflected that. Not necessary, but I think it would make for some interesting theory building.

I think the dual identity addition to H10 is interesting.

Comments on the Quality of English Language

NA

Author Response

Thank you very much for the second review! These are valuable comments and we hope that addressing them made the paper more readable and complete.

Comment 1:

I am still concerned about the screening criteria and asymetrical sample, but agree with the authors that the findings are still a contribution in spite of this (although I would still like more discussion of why this limitation matters).

Response 1:

We added a suggestion for future research in the limitations section in line 744. We could use open-ended questions to query the reluctant participants regarding their reason - then we could use systematic and automatic qualitative analysis to get the emerging themes. It would not solve the problem of the small sample, but it would provide us with more meaningful insights.

Comment 2:

I still don’t think the qualitative elements seems rigorous enough to address the hypotheses. There is just a lack of text and methodological discussion here. I don’t think that should keep this from being published, but I would advise framing and discussing it as exploratory or taking it out. 

Response 2:

We agree that this form of analysis is less common than the standard thematic and coding analysis methods, but we still stand by the value of the qualitative analysis as data that substantially enriches the quantitative results (method text was slightly refined starting in line 252). In the case of the word clouds, the emerging theme of "strangeness" gives depth to why Israeli Jew participants were deterred by the idea of telerobotics. The comments on "normalization" with Palestinian participants, when combined with the numerical data provide a good reason for further research . The themes in hopes, concerns, and theatre plays of Israeli Jews align with the needs model.

Comment 3:

H3 is a null hypothesis rather than a research hypothesis. Since the null hypothesis should not be interpreted I would suggest instead framing it as a RQ and not overinterpreting any lack of difference.

Response 3: We removed H3 and instead referred to RQ1 (line 108).

Comment 4:

H6 should be framed in terms of preferences rather than use (unless I am misunderstanding something). I also don’t know why the hypothesis is the way it is given a general propensity to want to dehumanize outgroups (especially when in heightened states of conflict).

Response 4:

We revised the hypothesis to the preference context. Also clarified that in this case we hypothesize that the ingroup telerobotic operator would prefer to appear themselves human when speaking to a member of the outgroup. We did not ask in the survey how the participants would like to see the outgroup represented, only how they would like to represent themselves.

Comment 5:

Leading into H7, the group identity models should be unpacked.

H8 the relation of salience to prejudice reduction should probably be discussed either here or in the discussion.

the disconnect between H7 and H8, i.e. between interpersonal and group level anonymity.

Response 5: We unpacked, revised and re-ordered the four hypotheses (now H5-H8) starting line 142. First, we discuss and hypothesize on the "positive" aspect of group identity expression - expression of nationality in mediated communication. Then we discuss the "negative" aspect - refraining from illegitimate stereotypes: appearing dehumanized, and then we discuss and hypothesize on the protective and loosening effect of anonymity.

Round 3

Reviewer 1 Report

Comments and Suggestions for Authors

I still have problems with the comparison of a representative and a non-representative sample and to take this as basis of comparison between two populations.

Minor: the dfs on line 323 arre wrong

Author Response

Comment 1:

"The dfs on line 323 are wrong"

Response 1:

Thank you for spotting that. We removed the misplaced numbers and the df is now t(615).